# LDMGAN: REDUCING MODE COLLAPSE IN GANS WITH LATENT DISTRIBUTION MATCHING

## ABSTRACT

Generative Adversarial Networks (GANs) have shown impressive results in modeling distributions over complicated manifolds such as those of natural images. However, GANs often suffer from mode collapse, which means they are prone to characterize only a single or a few modes of the data distribution. In order to address this problem, we propose a novel framework called LDMGAN. We first introduce Latent Distribution Matching (LDM) constraint which regularizes the generator by aligning distribution of generated samples with that of real samples in latent space. To make use of such latent space, we propose a regularized AutoEncoder (AE) that maps the data distribution to prior distribution in encoded space. Extensive experiments on synthetic data and real world datasets show that our proposed framework significantly improves GAN's stability and diversity.

## 1 INTRODUCTION

Generative models (Smolensky, 1986; Salakhutdinov & Hinton, 2009; Hinton & Salakhutdinov, 2006; Hinton, 2007; Kingma & Welling, 2013; Rezende et al., 2014; Goodfellow et al., 2014) provide powerful tools for unsupervised learning of probability distributions over difficult manifolds such as those of natural images. Among these models, instead of requiring explicit parametric specification of the model distribution and a likelihood function, Generative Adversarial Networks (GAN) (Goodfellow et al., 2014) only have a generating procedure. They generate samples that are sharp and compelling, which have gained great successes on image generation tasks (Denton et al., 2015; Radford et al., 2015; Karras et al., 2017; Zhang et al., 2018) recently. GANs are composed of two types of deep neural networks that compete with each other: a generator and a discriminator. The generator tries to map noise sampled from simple prior distribution which is usually a multivariate gaussian to data space with the aim of fooling the discriminator, while the discriminator learns to determine whether a sample comes from the real dataset or generated samples.

In practice, however, GANs are fragile and in general notoriously hard to train. On the one hand, they are sensitive to architectures and hyper-parameters (Goodfellow et al., 2014). For example, the imbalance between discriminator and generator capacities often leads to convergence issues. On the other hand, there is a common failure issue in GANs called mode collapse. The generator tends to produce only a single sample or a few very similar samples in that case, which means GANs put large volumes of probability mass onto a few modes.

We conjecture the mode missing issue in GANs is probably because GANs lack a regularization term that can lead the generator to produce diverse samples. To remedy this problem, in this work, we first propose a regularization constraint called Latent Distribution Matching. It suppresses the mode collapse issue in GANs by aligning the distributions between true data and generated data in encoded space. To obtain such encoded space, we introduce a regularized autoencoder which maps data distribution to a simple prior distribution, eg., a gaussian. As shown in Figure 1, we collapse the decoder of the regularized AE and generator of GAN into one and propose LDMGAN. Our framework can stabilize GAN's training and reduce mode collapse issue in GANs. Compared to other AE-based methods on 2D synthetic, MNIST, Stacked-MNIST, CIFAR-10 and CelebA datasets, our method obtains better stability, diversity and competitive standard scores.

## 2 BACKGROUND

GANs were initially proposed by Goodfellow et al. (2014). They contain two neural networks: a generator and a discriminator. Let $\{x_i\}_{i=1}^N$ denote the training data, where each $x_i \in \mathbb{R}^D$ is drawn from unknown data distribution $p_d(x)$. Generator is a neural network $G_\gamma$ that maps the noise vector $z \in \mathbb{R}^K$, typically drawn from a multi-variate gaussian, to data space $\tilde{x} \in \mathbb{R}^D$. The discriminator $D_\omega$ is a classification network that distinguishes real samples from generated samples. The parameters of these networks are optimized by solving the following minmax game:

$$\min_\gamma \max_\omega \mathcal{V}(G, D) = \mathbb{E}_x \log D(x) + \mathbb{E}_z \log(1 - D(G(z))) \tag{1}$$

where $\mathbb{E}_x$ indicates an expectation over the data distribution $p_d(x)$ and $\mathbb{E}_z$ indicates an expectation over the prior noise distribution $p(z)$. Given generator $G$, the optimal discriminator $D$ is $D_G^*(x) = \frac{p_d(x)}{p_d(x) + p_g(x)}$, where $p_g(x)$ is the model distribution. With the help of the optimal discriminator, the GAN objective is actually minimizing the Jensen Shannon divergence between the model distribution and the data distribution. And the global minimum is achieved if and only if $p_g(x) = p_d(x)$.

## 3 RELATED WORKS

There are many works that try to stabilize GAN's training and alleviate mode collapse in GANs. DCGAN (Radford et al., 2015) designed a class of deep convolutional architectures which has become standard architecture for training GANs. Improved GANs (Salimans et al., 2016) proposed several techniques like feature matching, mini-batch discrimination and historical averaging which stabilized GAN's training and reduced mode collapse. Unrolled GAN (Metz et al., 2016) proposed unrolling optimization of the discriminator objective to train generator. TTUR (Heusel et al., 2017) introduced two time-scale update rule and proved its convergence to a local Nash equilibrium. Arjovsky et al. (2017) analyzed the convergence properties of GANs and proposed WGAN which leveraged the Wasserstein distance and demonstrated its better convergence than Jensen Shannon divergence. However, WGAN required that the discriminator must lie on the space of 1-Lipschitz functions. It used a weight clipping trick to enforce that constraint. WGAN-GP stabilized WGAN by alternating the weight clipping by penalizing the gradient norm of the interpolated samples. SN-GAN (Miyato et al., 2018) proposed spectral normalization which controls the Lipschitz constraint of netowork layers.

Some works make efforts towards integrating AEs into GANs. AEs can be used in discriminators. EBGAN (Zhao et al., 2016) employed an autoencoder structure in its discriminator and introduced repelling regularization to prevent mode collapse. BEGAN (Berthelot et al., 2017) extended EBGAN by optimizing Wasserstein distance between AE loss distributions of real and generated samples. MDGAN (Che et al., 2016) contained a manifold step and a diffusion step that combined a plain autoencoder with GANs to suppress mode missing problem. However, MDGAN required two discriminators and it did not impose regularization on its AE. VAEGAN (Larsen et al., 2015) unified VAE and GAN into one model, and utilized feature-wise distance in intermediate features learned by discriminator to replace similarity metric in data space in VAE to avoid blur. AAE (Makhzani et al., 2015) used adversarial learning in its encoded space to perform variational inference. BiGAN (Donahue et al., 2016) and Adversarial Learned Inference (ALI) (Dumoulin et al., 2016) jointly trained an inference model and a generative model through adversarial game. VEEGAN (Srivastava et al., 2017) employed a reconstructor network to both map the true data distribution $p_d(x)$ to a gaussian and to approximately invert the generator network. AGE (Ulyanov et al., 2018) performed adversarial game between a generator and an encoder, and proposed to align the model distribution with data distribution in encoded space. They eliminated discriminator in their framework and used an encoder to extract statistics to align model distribution with data distribution.

The most related works to ours are VAEGAN and VEEGAN. VAEGAN's motivation was to combine VAE with GAN, and they relied on an ELBO, namely $\mathbb{E}_{q(z|x)}[\log p(x|z)] - D_{\mathrm{KL}}(q(z|x)\|p(z))$. While we perform marginal distribution matching, namely $\mathbb{E}_{q(z|x)}[\log p(x|z)] - D_{\mathrm{KL}}(q(z)\|p(z))$, and view the second term as a regularization term. Therefore there is no need of re-parameterization trick in our framework. Also we compare the distributions of the real samples and generated samples in the encoded space to suppress mode collapse in GANs which is different from VAEGAN.

VEEGAN had a reconstructor network which mapped the data distribution to a gaussian as well. However, VEEGAN autoencoded noise vectors rather than data items, and its training objective was formed as an adversarial game in joint space $(x, z)$ like BiGAN and ALI. Our regularized autoencoder is similar to AAE. But instead of using adversarial learning in encoded space which also has possibility of falling into mode collapse as in data space, we use explicit divergence to align aggregated posterior with the prior distribution.

## 4 METHOD

GAN's training can be viewed as an adversarial game between two players, in which the discriminator tries to distinguish between real and generated samples, while the generator tries to fool the discriminator by pushing the generated samples towards the direction of higher discrimination values. However, there are many points in data manifold that have high discrimination values. When collapse to a single mode is imminent, the gradient of the discriminator may point in similar directions for many similar points (Salimans et al., 2016). Since the discriminator iteself processes each sample independently, there is no mechanism to tell the outputs of the generator to become more dissimilar to each other. In this paper, we propose a framework called LDMGAN to address this problem. See Figure 1 for a glance of our model's architecture. Instead of exposing the dissimilarity of generated samples to the discriminator (Salimans et al., 2016; Zhao et al., 2016), we introduce a regularized autoencoder in our framework. We perform data dimensionality reduction on the training data to obtain a well-shaped manifold in encoded space, e.g., a factorized gaussian, by learning a regularized autoencoder. Then when sampling noise from a factorized gaussian to feed into the generator to produce generated samples, we encode the generated samples to latent space to enforce them to be coincident with the factorized gaussian. In this way, the generator receives a signal from that constraint to tell it to produce diverse samples. We call this constraint Latent Distribution Matching (LDM). In the following, we describe LDM and the regularized autoencoder in detail.

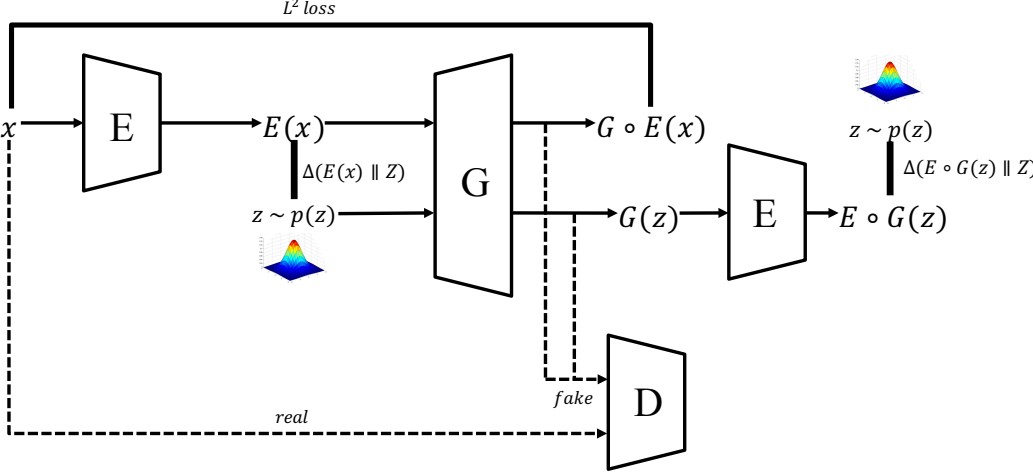

Figure 1: Our LDMGAN's architecture. The thick lines denote added losses to GANs.

### 4.1 LATENT DISTRIBUTION MATCHING

The main idea of LDM is to introduce an encoder function $F_\theta$ that can map the true data distribution $p(x)$ to a prior Gaussian $p(z)$. To understand why such an encoder helps alleviate address mode collapse, consider the example in Figure 2. The middle vertical panel represents the data space, where in this example the true data distribution $p_d(x)$ is a mixture of two Gaussians. The bottom panel depicts the input to the generator (latent code space), which is drawn from $p(z)$, and the top panel depicts the results of applying the encoder network to the generated data and true data. Assume that the encoder approximately maps the data distribution to the prior gaussian (see $q(z)$

above the top panel in Figure 2). The purple arrows above the middle panel show the action of the encoder on the true data, whereas the green arrows show the same action on the generated data. If mode collapse occurs as it does in this example that the generator $G_\gamma$ captures only one of the two modes of $p_d(\boldsymbol{x})$, it would cause distribution mismatch between the two marginal distributions $q(\boldsymbol{z})$ and $q'(\boldsymbol{z})$. The definitions of $q(\boldsymbol{z})$ and $q'(\boldsymbol{z})$ go as follow:

$$q(\boldsymbol{z}) = \int_{\boldsymbol{x}} q(\boldsymbol{z}|\boldsymbol{x})p_d(\boldsymbol{x})d\boldsymbol{x} \tag{2}$$

$$q'(\boldsymbol{z}) = \int_{\boldsymbol{x}} q(\boldsymbol{z}|\boldsymbol{x})p_g(\boldsymbol{x})d\boldsymbol{x} \tag{3}$$

where $p_g(\boldsymbol{x})$ denotes the model distribution of $G_\gamma$ in Equation 3. This mismatch is evaluated by a divergence measure $\Delta(q'(\boldsymbol{z})\|q(\boldsymbol{z}))$ in the latent space. We only require this divergence to be non-negative and zero if and only if the distributions are identical $(\Delta(q'(\boldsymbol{z})\|q(\boldsymbol{z})) = 0 \iff q'(\boldsymbol{z}) = q(\boldsymbol{z}))$. Therefore, this mismatch value function is defined as:

$$\mathcal{V}^1_{mismatch} = \Delta(q'(\boldsymbol{z})\|q(\boldsymbol{z})) \tag{4}$$

However, comparing two empirical (hence non-parametric) distributions $q'(\boldsymbol{z})$ and $q(\boldsymbol{z})$ is difficult. We avoid this issue by introducing an intermediate reference distribution which is also the prior distribution $p(z)$, resulting in:

$$\mathcal{V}^2_{mismatch} = \Delta(q'(\boldsymbol{z})\|p(\boldsymbol{z})) - \Delta(q(\boldsymbol{z})\|p(\boldsymbol{z})) \tag{5}$$

As we have assumed that the encoder network can approximately map the data distribution to a gaussian, we can get rid of $\Delta(q(\boldsymbol{z})\|p(\boldsymbol{z}))$ with minor error. Finally, this mismatch term turns out to be

$$\mathcal{V}^3_{mismatch} = \Delta(q'(\boldsymbol{z})\|p(\boldsymbol{z})) \tag{6}$$

It can be used to detect mode collapse in generator. We call this mismatch term as Latent Distribution Matching (LDM), and use this term to regularize the generator in GANs to combat mode collapse.

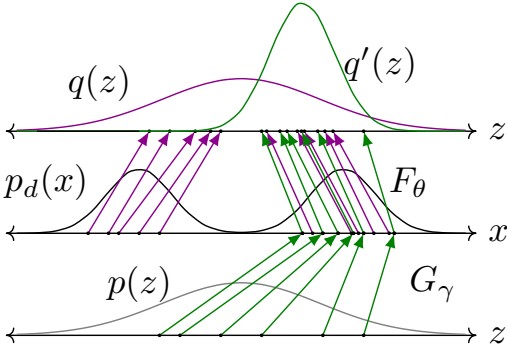

Figure 2: An example of how LDM helps to address mode collapse. See text for explaination.

## 4.2 REGULARIZED AUTOENCODER

AutoEncoders have long been used for data dimensionality reduction and feature learning. We impose the regularization that the distribution in the encoded space of real data is coincident with the prior distribution $p(\boldsymbol{z})$ on the autoencoder. We call this autoencoder the regularized autoencoer in this paper. There are two ways of utilizing the regularized autoencoder. One is to pretrain a regularized autoencoder, then use the encoder of the regularized autoencoder only. The other is to combine

the autoencoder training with GAN objective which has already been proposed in many GAN variants (Larsen et al., 2015; Che et al., 2016; Srivastava et al., 2017). We combine autoencoder with GANs in our model for three reasons: (1) to guarantee the learning is grounded on all training samples; (2) to regularize the generator in producing samples to resemble real ones; (3) last but not least, to obtain an encoder that can approximately map the data distribution to a gaussian. In the following, we will explain the reasons for combining autoencoders in our model and describe our framework formally.

### 4.2.1 REASONS OF COMBINING AUTOENCODERS WITH GANs

One of the reasons that mode collapse occurs is probably that the areas near the missing modes are rarely visited by the generator, therefore providing very few samples to improve the generator around those areas. And only when $G(z)$ is very close to the missing modes can the generator get gradients to push itself towards the missing modes. Consider a model combining GANs with reconstruction loss in data space $d(x, G(E(x)))$. Given enough data examples $x$ available in training data for minor mode $M_0$, we know $G \circ E$ is a good autoencoder, then $G \circ E(x)$ will be located very close to the minor mode $M_0$. Therefore the gradient from the discriminator $\nabla \log D(G \circ E(x))$ might be able to push these points towards the minor mode $M_0$ and the learning is grounded on all training samples. For that purpose, the following loss is added to the generator:

$$\mathcal{L}_G = \mathbb{E}_x[\lambda_1 d(x, G \circ E(x)) - \log(D(G \circ E(x)))] \tag{7}$$

As the regularized autoencoder combined with GAN has imposed gaussian on the encoded space of real data samples, this guarantees such $E(x)$ for minor modes can be sampled with fair possibility when sampling $z$ from gaussian that are fed to the generator. In addition, as lots of image synthesis works (Isola et al., 2017; Zhu et al., 2017) have used $L^1$ or $L^2$ loss in data space and achieved successes in producing realistic images. We use the autoencoder loss $d(x, G \circ E(x))$ in data space as well for the purpose of regularizing the generator in producing samples to resemble real ones.

### 4.2.2 REGULARIZATION ON THE ENCODED SPACE

In this section, we formally demonstrate the regularized autoencoder we combine with GANs. Let $x$ be the data sample vector, $\tilde{x}$ be the reconstructed sample vector of $x$ by the autoencoder and $z$ be the latent code vector of the autoencoder. Let $p(z)$ be the prior distribution we want to impose on the codes, $q(z|x)$ be an encoding distribution and $p(x|z)$ be the decoding distribution. Also let $p_d(x)$ be the data distribution, and $p_g(x)$ be the model distribution. We desire an autoencoder whose aggregated posterior $q(z)$ is coincident with a prior distribution $p(z)$. In this paper, we use a factorized gaussian as the prior distribution. The encoding function of the autoencoder $q(z|x)$ defines the aggregated posterior in Equation 2. We choose a deterministic function form for $q(z|x)$ which means the stochasticity of $q(z)$ comes only from the the data distribution $p_d(x)$. We impose a regularization term on the encoded space of the autoencoder to minimize: $\Delta(q(z)\|p(z))$. The autoencoder, meanwhile, attempts to minimize the reconstruction error $d(x, \tilde{x})$ (here $d$ is some similarity metric, besides $L^1$ or $L^2$ distance in data space, it can also be distance metric in the feature-wise space like the intermediate layers of well-known VGG (Simonyan & Zisserman, 2014) network or the discriminator of GANs).

Therefore,the regularized autoencoder is trained in two phases: the *reconstruction* phase and the *regularization* phase. In the *reconstruction* phase, the regularized autoencoder uses SGD to update the encoder and decoder's parameters to reconstruct the input data vector. In the *regularization* phase, the autoencoder uses SGD to update the encoder's parameters to match the aggregated posterior with the prior distribution. Implemented by deep neural networks, with arbitrary capacity, the universal approximation theorem (Hornik et al., 1989; Cybenko, 1989) guarantees the regularized autoencoder is capable of approximately matching the aggregated posterior $q(z)$ with the prior distribution $p(z)$.

### 4.3 TRAINING OBJECTIVE

Given the encoder $E_\theta$ which is capable of approximately mapping the data distribution to a gaussian, consider a standard GAN training loss plus the LDM (Latent Distribution Matching) constraint on the generator:

$$\min_{G_\gamma} \max_{D_\omega} \mathcal{V}(D, G) = \mathbb{E}_{\boldsymbol{x}} \log D(\boldsymbol{x}) + \mathbb{E}_{\boldsymbol{z}} \log(1 - D(G(\boldsymbol{z}))) + \Delta(q'(\boldsymbol{z}) \| p(\boldsymbol{z})) \tag{8}$$

This training objective Equation 8 is what we apply in GAN framework when not concerning combining the regularized autoencoder into GAN. And after unifying the regularized autoencoder and GAN described above, the objective becomes:

$$\min_{G_\gamma E_\theta} \max_{D_\omega} \mathcal{V}(D, G, E) = \mathbb{E}_{\boldsymbol{x}}[\log D(\boldsymbol{x}) + \log(1 - D(G \circ E(\boldsymbol{x})))] + \mathbb{E}_{\boldsymbol{z}} \log(1 - D(G(\boldsymbol{z}))) \\ + \lambda_1 d(\boldsymbol{x}, G \circ E(\boldsymbol{x}))] + \lambda_2 \Delta(q'(\boldsymbol{z}) \| p(\boldsymbol{z})) \tag{9}$$

where $\lambda_1$ and $\lambda_2$ are hyper-parameters that control the loss weights of reconstruction error and LDM constraint respectively. Note that the LDM term $\Delta(q'(z) \| p(z))$ is minimized only with respect to $G_\gamma$, which means the parameters of the encoder $E$ are fixed during backpropagating the LDM term. The divergence measure between the empirical distribution and the prior distribution in the latent $M$-dimensional space that we use in this paper goes as follow:

$$D_{\text{KL}}(Q \| \mathcal{N}(0, \boldsymbol{I})) = -\frac{1}{2}M - \sum_{i=1}^{M} \log \sigma_i + \sum_{i=1}^{M} \frac{\sigma_i^2 + \mu_i^2}{2} \tag{10}$$

where $\mu_i$ and $\sigma_i$ are the means and the standard deviations of the fitted empirical distribution $Q$ along various dimensions. The training algorithm and architecture of our proposed model is depicted in Algorithm 1 and Figure 1 respectively.

---

**Algorithm 1** Our LDM-GAN's training algorithm

---

1: $D_\omega, E_\theta, G_\gamma \leftarrow$ initialize network parameters respectively
2: **repeat**
3:      $X \leftarrow$ Random mini-batch of $m$ data points from dataset
4:      $Z \leftarrow$ Random $m$ samples from prior distribution $p(\boldsymbol{z})$
5:      $\tilde{Z} \leftarrow E_\theta(X)$
6:      $\tilde{X} \leftarrow G_\gamma(Z)$
7:      $X_{rec} \leftarrow G_\gamma(\tilde{Z})$
8:      $Z_{rec} \leftarrow E_\theta(\tilde{X})$
9:      // *Training discriminator D*
10:      $g_\omega \leftarrow -\nabla_\omega[\log D_\omega(X) + \log(1 - D_\omega(\tilde{X})) + \log(1 - D_\omega(X_{rec}))]$
11:      // *Training encoder E*
12:      $g_\theta \leftarrow \nabla_\theta d(X, X_{rec}) + D_{\text{KL}}(q(\tilde{Z}) \| p(\boldsymbol{z}))$
13:      // *Training generator G*
14:      $g_\gamma \leftarrow \nabla_\gamma \log(1 - D_\omega(\tilde{X})) + \log(1 - D_\omega(X_{rec})) + \lambda_1 d(X, X_{rec}) + \lambda_2 D_{\text{KL}}(q'(Z_{rec}) \| p(\boldsymbol{z}))$
15:      // *Using SGD to update model's parameters*
16:      $\omega \leftarrow \omega - \eta g_\omega; \theta \leftarrow \theta - \eta g_\theta; \gamma \leftarrow \gamma - \eta g_\gamma$
17: **until** *convergence*
18: **return** $D, E, G$

---

## 5 EXPERIMENTS

In this section, we compare our model with four different models on 2D Synthetic, MNIST, Stacked-MNSIT, CIFAR-10 and CelebA datasets. We do not finetune the hyper-parameters $\lambda_1$ and $\lambda_2$ in our model, and by default set $\lambda_1 = 1$ and $\lambda_2 = 1$.

| | 2D Grid | | 2D Ring | |
|---|---|---|---|---|
| | **Modes (Max 25)** | **% High Quality Samples** | **Modes (Max 8)** | **% High Quality Samples** |
| **GAN** | $3.4 \pm 0.70$ | $11.10 \pm 0.02$ | $1 \pm 0.00$ | $99.96 \pm 0.01$ |
| **BiGAN** | $20.4 \pm 2.18$ | $66.45 \pm 0.11$ | $6.9 \pm 0.93$ | $38.07 \pm 0.07$ |
| **VEEGAN** | $1 \pm 0.00$ | $99.45 \pm 0.01$ | $1 \pm 0.00$ | $99.89 \pm 0.01$ |
| **VAEGAN** | $8.9 \pm 0.33$ | $98.60 \pm 0.01$ | $1 \pm 0.00$ | $100.00 \pm 0.00$ |
| **LDMGAN** | $\mathbf{23.1} \pm 1.05$ | $65.36 \pm 0.07$ | $\mathbf{8} \pm 0.00$ | $97.86 \pm 0.01$ |

Table 1: Modes captured and percentage of high quality sample on 2D synthetic data. Our model consistently captures the highest number of modes and produces competing percentage of high quality samples.

## 5.1 SYNTHETIC 2D DATA

Evaluation of mode collapse in GANs is very difficult especially when trained on natural images. However, the missing modes can be calculated precisely when using synthetic data for training. In this section we compare five different models on two synthetic datasets: a mixture of eight 2D Gaussian distributions arranged in a ring, and a mixture of twenty-five 2D Gaussian distributions arranged in a grid. The covariance matrices and centroids have been chosen such that the distributions exhibit lots of modes separated by large low-probability regions, which make them decently hard tasks despite the 2D nature of the dataset.

All models share the same network architectures for fair comparison. The encoder and the generator network are consisted of two hidden layers, while the discriminator network only has one hidden layer. All networks are implemented by fully-connected layers and each layer has 64 neurons. The dimension of the prior input is 2. For VEEGAN and BiGAN, the discriminator's input is augumented by the size of the latent code. Note that the VAEGAN we used here is slightly different from the original paper, since the network architectures are shallow and the data is low-dimensional: we used $L^2$ loss on the data space for simplicity instead of on features learned by the discriminator.

To quantify the mode collapse behavior we report two metrics: the number of modes that each model captures and the percentage of high quality samples. We draw 2048 samples from each model for evaluation. The high quality samples are defined as the points which are within three standard deviations of the nearest mode, and if there are more than twenty such samples registered to a mode, the mode is considered as captured. The results are depicted in Table 1 and the numbers are averaged over eight runs. We also show the generated distribution of each model in Figure 3

As we can see from Table 1, our model captures the greatest number of modes on both synthetic datasets. The generator learned by our model is sharper and closer to the true distribution as shown in Figure 3. Though other models may obtain greater percentage of high quality samples, for example, vanilla GAN generates $99.96\%$ high quality samples on 2D Ring dataset. However, it runs into severe mode missing problem. Vanilla GAN only captures one mode on 2D Ring dataset.[1]. VEEGAN is able to capture most of the modes in some settings (LDMGAN also performs as well as VEEGAN under those settings), but it fails to generalize well under our settings. In particular, while our model resembles VAEGAN in some ways, VAEGAN suffers from some degree of mode missing problem as well and it is far from competitive compared to our proposed model.

## 5.2 EXHAUSTIVE GRID SEARCH ON MNIST

In order to systemically explore the effect of our proposed LDMGAN on alleviating mode collapse in GANs, we use a large scale grid search of different GAN hyper-parameters on the MNIST dataset. We use the same hyper-parameters settings for both vanilla GAN and our LDMGAN, and the encoder in our model is the "inverse" of the generator. The search range is listed in Table 2. Our grid search is similar to those proposed in Che et al. (2016); Zhao et al. (2016). Please refer to it for detailed explanations regarding these hyper-parameters.

---

[1]Note that under certain settings of parameters, vanilla GAN can also recover all modes on 2D Ring and 2D Grid datasets.

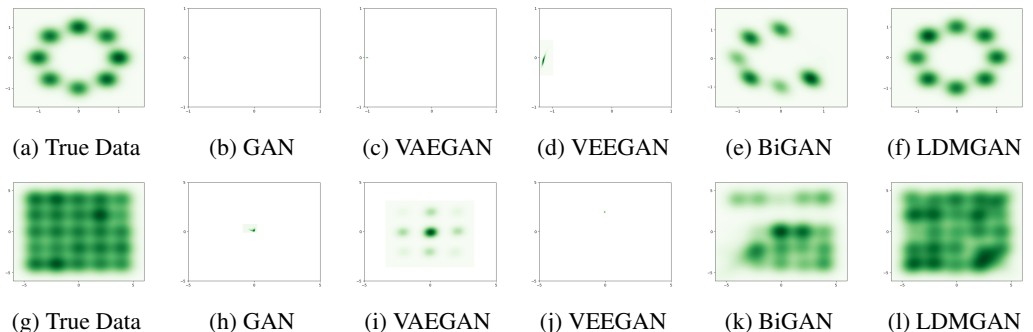

| (a) True Data | (b) GAN | (c) VAEGAN | (d) VEEGAN | (e) BiGAN | (f) LDMGAN |
| (g) True Data | (h) GAN | (i) VAEGAN | (j) VEEGAN | (k) BiGAN | (l) LDMGAN |

Figure 3: Density plots of the true data and generator distributions from different GAN methods trained on mixtures of Gaussians arranged in a ring (top) or a grid (bottom).

| | |
|---|---|
| nLayerG | [2,3,4] |
| nLayerD | [2,3,4] |
| sizeG | [400,800,1600,3200] |
| sizeD | [256,512,1024] |
| dropoutD | [True,False] |
| optimG | [SGD,Adam] |
| optimD | [SGD,Adam] |
| lr | [1e-2,1e-3,1e-4] |

Table 2: Grid Search for Hyper-parameters

In order to estimate both the missing modes and the sample qualities, we first train a regular CNN classifier on the MNIST digits, and then apply it to compute the MODE Score which is proposed in Che et al. (2016). MODE Score is a variant of Inception Score (IS)(Salimans et al., 2016) and its definition goes as follow:

$$\exp(\mathbb{E}_{\boldsymbol{x}} D_{\mathrm{KL}}(p(y|\boldsymbol{x})\|p(y)) - D_{\mathrm{KL}}(p^*(y\|p(y)))) \tag{11}$$

where $p(y|\boldsymbol{x})$ is the softmax output of a trained classifier of the labels, and $p^*(y)$ and $p(y)$ are the marginal distributions of the label of the generated samples and training samples respectively. We train each architecture for 50 epochs and after that draw 10K samples for evaluation. The resulting distribution of MODE Score is shown in Figure 4. It can be seen that our model clearly improves the sample quality and diversity compared to GANs.

### 5.3 STACKED MNIST

Following Metz et al. (2016), we evaluate our method on the stacked MNIST dataset, a variant of the MNIST data specifically designed to increase the number of discrete modes. The data is synthesized by stacking three randomly sampled MNIST digits along the color channel resulting in a $28 \times 28 \times 3$ image. We now expect 1000 modes on this dataset, corresponding to the number of possible triples of digits.

We generate a dataset of 50000 images for training. And for fair comparation we use the same architecture for all models. The generator and discriminator networks are implemented following DCGAN. For encoder network, we use a simple two layer MLP network without any regularization layers. For BiGAN and VEEGAN, we concatenate the last convolutional layer of the discriminator with the latent code which follows by two fully-connected layers in order to perform joint space discrimination. The dimension of the prior input is 64. Each model is trained for 50 epochs.

As the true locations of the modes in this data are unknown, the number of modes are estimated using a trained classifier as described in subsection 5.2. We draw 26000 samples from each model for

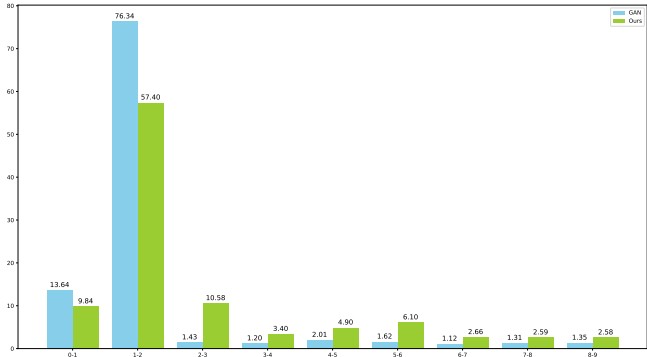

Figure 4: The distribution of MODE scores for GAN and our LDM-GAN. Higher MODE Score is better.

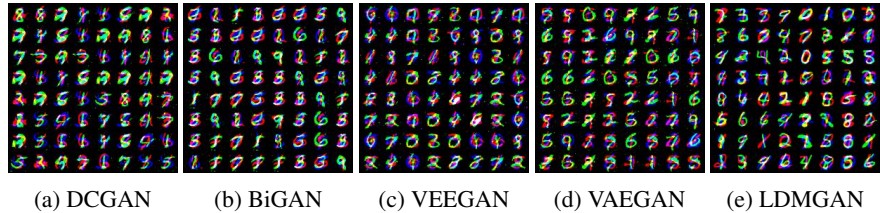

(a) DCGAN          (b) BiGAN          (c) VEEGAN          (d) VAEGAN          (e) LDMGAN

Figure 5: Random samples of different methods on Stacked-MNIST. Zoom in to see details.

evaluation. As a measure of quality, following Metz et al. (2016), we also report the KL divergence between the model distribution and the data distribution. The results are averaged over 5 runs. As reported in Table 3, our model recovers greatest number of modes and better matches the data distribution than any other methods.

|  | Modes (Max 1000) | KL Divergence |
|---|---|---|
| **DCGAN** | $165.0 \pm 49.06$ | $3.36 \pm 0.21$ |
| **BiGAN** | $135.6 \pm 89.64$ | $3.22 \pm 0.23$ |
| **VEEGAN** | $178.8 \pm 48.07$ | $3.35 \pm 0.32$ |
| **VAEGAN** | $206.4 \pm 32.62$ | $3.21 \pm 0.34$ |
| **LDMGAN** | $\mathbf{988.8 \pm 2.40}$ | $\mathbf{0.15 \pm 0.06}$ |

Table 3: Degree of mode collapse, measured by modes captured and sample quality (as measured by KL) on Stacked-MNIST. Our model captures the most modes and also achieves the highest quality.

## 5.4 CIFAR-10 AND CELEBA

For CIFAR-10 and CelebA datasets, we measure the performance with FID scores (Heusel et al., 2017). FID can detect intra-class mode dropping, and measure the diversity as well as the quality of generated samples. FID is computed from 10K generated sampels and the pre-calculated statistics on all training data provided in the code repository[2]. Our default parameters are used for all experiments. And the model architecture is the same as the standard DCGAN. We train each model for 50 epochs on CIFAR-10 and 30 epochs on CelebA. The dimension of the prior input is 100. All of our experiments are conducted using the unsupervised setting. We evaluate the FID score for each model per epoch during training, and leave the best one per run. As we can not reproduce VEEGAN and BiGAN on these datasets using the same architecture, we do not report FID scores on these two models.

---

[2]https://github.com/bioinf-jku/TTUR

The reported results are depicted in Table 4 and are averaged over 5 runs. It is important to mention that we do not pursue the state-of-the-art FID scores on both datasets as we only want to prove that our model improves GANs. It can be seen that our model has both the best FID scores and averaged FID scores on CIFAR-10 and CelebA. The improvement on CelebA is smaller compared to that on CIFAR-10. We suspect that this is due to the less complicated structure of face images, but anyway our model still consistently performs better than DCGAN and VAEGAN. The random samples generated from different models on both datasets are shown in Appendix A. Finally, to show that our proposed model are not just simply memorizing the training data, we interpolate between random vectors on generator of our model trained on CelebA dataset and the results are depicted in Appendix A.

| | CIFAR-10 | | CelebA | |
|---|---|---|---|---|
| | **best FID** | **FID** | **best FID** | **FID** |
| **DCGAN** | 51.19 | $51.79 \pm 0.38$ | 14.28 | $14.82 \pm 0.88$ |
| **VAEGAN** | 49.50 | $50.82 \pm 0.71$ | 13.88 | $14.98 \pm 0.77$ |
| **LDMGAN** | **44.87** | **47.02** $\pm 1.18$ | **12.79** | **13.32** $\pm 0.45$ |

Table 4: FID scores for different models, lower is better. Best FID is the best run in 5 trial.

## 6 CONLUSION

We propose a robust AE-based GAN model with novel Latent Distribution Matching constraint, called LDM-GAN, that can address the mode collapse effectively. Our model is different from previous works: (1) We propose a novel regularization constraint called Latent Distribution Matching to align the distributions of real data and generated data in encoded space. (2) We propose a new regularized autoencoder for approximately mapping data distribution to a gaussian. (3) We designed a novel AE-based GAN that drastically alleviate mode collapse in GANs. Extensive experiments demonstrate that our method is stable and does not suffer from mode collapse problem in Synthetic 2D, MNIST, Stacked-MNIST datasets. Furthermore, we achieve better FID scores on CIFAR-10 and CelebA datasets compared to baseline models without any other tricks and techniques. These demonstrate that our proposed LDMGAN indeed improves GAN's stability and diversity.

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

# A   APPENDIX

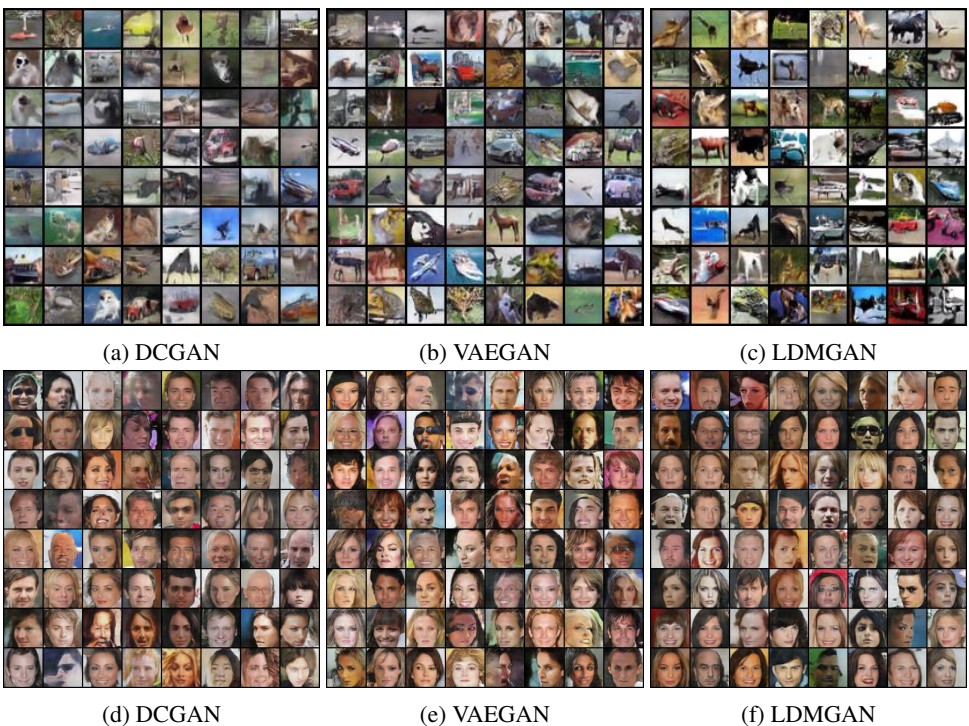

Figure 6: Random samples of different methods. Zoom in to see details.

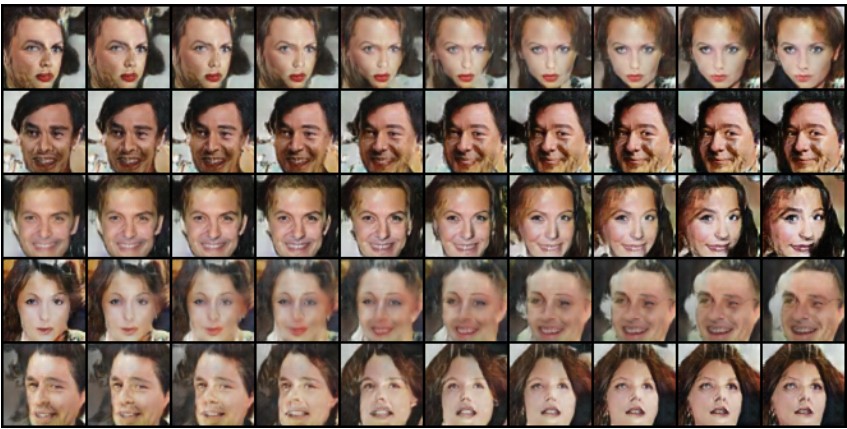

Figure 7: Generated samples by interpolating between two different vectors (from left to right).

