# OpenReview forum: "LDMGAN: Reducing Mode Collapse in GANs with Latent Distribution Matching"
_ICLR.cc/2020/Conference — Reject_

### Official Review · AnonReviewer1 · 2019-10-22
**Official Blind Review #1**

**Rating:** 1

**Review:**

LDMGAN: Reducing Mode Collapse in GANs with Latent Distribution Matching

Summary:

This paper proposes a modification to the VAE-GAN model where mode coverage is encouraged by passing samples G(Z) through the encoder and minimizing the forward KL between E(G(Z)) and the prior over Z. Results are presented on synthetic MoG datasets, MNIST variants, CIFAR and CelebA.

My Take:

This paper’s only point of novelty over a vanilla VAE-GAN implementation is the inclusion of the KL(E(G(Z)) || p) term in the generator loss, which is very similar to the idea behind VEEGAN. The relative novelty over VEEGAN is also limited, the description of the method is exceptionally similar to the description used in the VEEGAN paper (going so far as to copy-paste a figure straight from VEEGAN without attribution), and the comparison to VEEGAN in the related work section is not sufficiently fleshed out. The difference in performance over a vanilla VAE-GAN on CIFAR and CelebA is negligible (6-9% relative reduction in FID on an already weak baseline), so there is no compelling empirical reason to adopt this method. I argue strongly in favor of rejection.

Notes:

-Mode collapse (when many points in z map to an unexpectedly small region in G(z)) is a different phenomenon from mode dropping (when many points in x are not represented in G(z), i.e. no point in z maps to a cluster of x’s, as is the case if e.g. a celebA model generates frowning and neutral faces but no smiling faces). While these phenomena often co-occur (especially during complete training collapse), they are not the same thing, and this paper conflates them throughout.

-For the synthetic dataset examples the comparison against VEEGAN appears to be unfair—it’s one thing to report robustness to hyperparameters, but this seems more like the authors have intentionally picked settings for which VEEGAN happens to fail (by halving the width). If the authors wish to use settings different from the standard ones used by most other papers which test on the MoG datasets they should justify it thoroughly and include this justification in their analysis—do we have a compelling reason to believe that the LDM method is better suited to learning lower capacity models in general?

Minor:

Typos throughout, like “regularized autoencoer.” Please thoroughly proofread your paper for grammar and spelling mistakes.

There are formatting errors in the PDF, such as at the top of page 8. Please examine your paper for formatting mistakes.



**Experience Assessment:**

I have published in this field for several years.

**Review Assessment: Checking Correctness Of Derivations And Theory:**

I carefully checked the derivations and theory.

**Review Assessment: Checking Correctness Of Experiments:**

I carefully checked the experiments.

**Review Assessment: Thoroughness In Paper Reading:**

I read the paper thoroughly.

---

### Official Review · AnonReviewer3 · 2019-10-24
**Official Blind Review #3**

**Rating:** 3

**Review:**

It is argued in this paper that GANs often suffer from mode collapse, which means they are prone to characterize only a single or a few modes of the data distribution. In order to address this problem, the paper proposed a framework called LDMGAN which constrains the generator to align distribution of generated samples with that of real samples in latent space by introducing a regularized AutoEncoder that maps the data distribution to prior distribution in encoded space.

The major difference of this paper from many traditional GANs is to constrain the distributions of generated data same as distributions of true data in latent space instead of constrain the ability of discriminator. The authors detailed their motivation, the algorithm, and also reported a series of evaluation results on several datasets.

Generally, this paper was well written. However, this paper has the following major concerns:
	（1） Though somewhat new, the novelty of this paper may be incremental to me. It looks like a combination of VEEGAN and AAE.  Though the authors mentioned that VEEGAN autoencoded the noise vectors rather than data items, and AAE exploited the adversarial learning in the encoded space rather than using an explicit divergence,   it appears not significant to me between the proposed model and these two models. At least the authors did not  address sufficiently how significant the proposed method would be.

	 （2） The paper tested the proposed algorithm with a 2D Synthetic dataset. However, I found a lot of discrepancies in the results presented in Table 1 with other published works. The authors show 1 of mode captured on 2D Grid and 2D Ring using the VEEGAN method. However, the VEEGAN paper shows they get 24.6 and 8 on 2D Ring and 2D Grid respectively. Such discrepancies were also observed in Figure 3. These discrepancies must be explained.

	（3） In Figure 4, the authors showed the distribution of MODE scores for GAN and LDMGAN. From the figure, it seemed that LDMGAN improved the sample quality and diversity compared to GANs, but it is still prone to characterizing only a single or a few modes of the data distribution. In another word,  this may alleviate the problem but may not fully solve the problem.  Another minor point, the coordinate and legend are too small in this figure. It would be better if they become bigger.

	（4） The results of Table 4 is not convincing because the comparative methods are truly out-of-date. It would be more convincing if more latest methods can be compared with LDMGAN method. Those results reported are far lower than the state-of-the-art performance in these datasets.

         (5) There is a mistake in the second term of equation 11, it should be ⋯〖-D〗_KL (p^* (y)||p(y))  )


**Experience Assessment:**

I have published one or two papers in this area.

**Review Assessment: Checking Correctness Of Derivations And Theory:**

I assessed the sensibility of the derivations and theory.

**Review Assessment: Checking Correctness Of Experiments:**

I carefully checked the experiments.

**Review Assessment: Thoroughness In Paper Reading:**

I read the paper thoroughly.

---

### Official Review · AnonReviewer2 · 2019-10-28
**Official Blind Review #2**

**Rating:** 1

**Review:**

The manuscript contains typing and grammatical errors. I also think that the presentation is not clear enough. The notation is not clearly defined also. I hope I am not missing it, but the function combination operation is not defined (e.g. f(x) o g(x), to denote f(g(x)), and used a lot.

Just under equation 4, you are saying that comparing empirical distributions is difficult, and hence propose an alternative objective. But you could actually compare the two empirical distributions with a sample based divergence measure such maximum mean discrepancy, or Frechet Inception distance. Why don't you do this?

In the algorithms section, I don't understand why you don't compare with widely used WGAN-GP model. DCGAN, and VAEGAN are relatively old models. Your only results in Celeb-A are compared against these relatively older models. Also by looking at the provided samples, I think the quality of the results are far away from what is obtained with state of the Generative models currently.

Overall, I think the idea is sound and sensible, but I don't think that results are convincing enough for a conference paper. Furthermore, the writing needs to be improved.

minor comments:

In Figure 2, I think you should provide the explanations also in the caption to make the concept easier to follow for the reader.

In Algorithm 1, I think it would be easier for the reader to follow if you don't use shorthand notations defined in lines 5, 6, 7, 8.


**Experience Assessment:**

I have published one or two papers in this area.

**Review Assessment: Checking Correctness Of Derivations And Theory:**

I assessed the sensibility of the derivations and theory.

**Review Assessment: Checking Correctness Of Experiments:**

I assessed the sensibility of the experiments.

**Review Assessment: Thoroughness In Paper Reading:**

I read the paper at least twice and used my best judgement in assessing the paper.

---

### Decision · Program_Chairs · 2019-12-19

**Decision:**

Reject

**Comment:**

This paper proposes to mitigate mode collapse in GANs by encouraging distribution matching in the latent space. Reviewers 1 and 3 expressed concerns that the methodology is too incremental in the context of the existing literature (VEEGAN, VAE-GAN, AAE). This, combined with the lack of up-to-date baselines, makes it difficult to access the significance of the proposed modifications. The quality and precision of the writing can also be improved to meet the standards of publication at a top-tier conference.